

# Zircon luminescence dating revisited

Christoph Schmidt[1], Théo Halter[1], Paul R. Hanson[2], Alexey Ulianov[3], Benita Putlitz[3], Georgina E. King[1], and Sebastian Kreutzer[4]

[1]Institute of Earth Surface Dynamics, University of Lausanne, Lausanne, 1015, Switzerland
[2]Conservation and Survey Division, School of Natural Resources, University of Lincoln-Nebraska, Lincoln, 68583, United States of America
[3]Institute of Earth Sciences, University of Lausanne, Lausanne, 1015, Switzerland
[4]Institute of Geography, Heidelberg University, 69120 Heidelberg, Germany

*Correspondence to*: Christoph Schmidt (christoph.schmidt@unil.ch)

**Abstract.** Luminescence dating plays a pivotal role in Quaternary science, yet ongoing methodological challenges persist in refining the temporal range, accuracy, and precision of luminescence methods. Our contribution revisits zircons as potential alternative dosimeters to quartz, feldspar, or calcite for routine dating applications. The essential advantage of zircons over other minerals is the time-invariant and high internal dose rate due to high radionuclide contents, dominating over the more challenging-to-assess external contribution. Reported drawbacks are low zircon abundance, laborious sample preparation, signal instabilities, unknown optical signal resetting rates, and low signal intensities. Our present study uses modern luminescence detection equipment and analytical methods to investigate mineral separation, mineral characteristics, bleachability, signal spectra and intensities as well as the potential to auto-regenerate signals. We present results for two zircon samples different in provenance, trace element composition and luminescence characteristics, each of them containing a couple of hundred grains. Optically stimulated luminescence (OSL) signal resetting rates of zircon in response to simulated sunlight exposure are orders of magnitudes faster than for feldspar and slightly slower than for quartz. The recorded thermoluminescence (TL) spectra confirm previously published results with luminescence emissions in the UV/violet and red wavelength range, supplemented by narrowband emissions associated with rare earth element dopants. Storage experiments of single zircon grains for auto-regenerated measurements over 1.5 years yielded very low OSL signals. At the same time, after only three weeks, we measured acceptable TL signal intensities at the cost of lower bleaching rates. To date, the auto-regeneration approach seems to be a promising and accurate approach to date zircon light exposure events, especially when combining the natural OSL with auto-regenerated TL. However, further studies are required to optimise signal intensities and establish zircons as viable targets for routine dating applications.



## 1 Introduction

Luminescence dating of natural quartz and feldspar minerals is one of the most important and influential geochronological techniques in geosciences and archaeology on Quaternary timescales. Direct dating of the mineral's last sunlight exposure (via optically stimulated luminescence, OSL; Huntley et al., 1985) or heating (via thermoluminescence, TL; Aitken, 1985; Mercier et al., 1995) allows for temporal constraints of geomorphological events and processes and of archaeological site formation and human action. However, in best-case scenarios, the method's full age uncertainty is 4–5 % but normally ranges from 8–

10% (Rhodes, 2011; Wallinga and Cunningham, 2015; Lamothe, 2016; Lancaster et al., 2016). These values are appreciably greater than uncertainties associated with radiometric techniques (e.g., radiocarbon dating; Hajdas et al., 2021) or methods of incremental counting (varve chronology, dendrochronology; Vandergoes et al., 2018; Zarczyński et al., 2018). Consequently, such uncertainty margins sometimes complicate the unequivocal interpretation of luminescence data with regard to the research question. For instance, OSL ages of loess deposits do not generate the temporal resolution necessary to assign millennial-scale

climate fluctuations (e.g., Greenland Interstadials; Moine et al., 2017; Fischer et al., 2021), and Holocene OSL chronologies of dune re-activation phases as a proxy for droughts in the Great Plains (USA) are not precise enough to be reconciled with multi-year climate forcing mechanisms (Forman et al., 2001; Buckland et al., 2019). In archaeology, the uncertainty assigned to luminescence ages often hinders the unambiguous temporal separation of lithic techno-complexes (Guerin et al., 2013).

Since the early 1970s, zircon (tetr. $ZrSiO_4$) was identified as a potential TL dosimeter (Zimmerman, 1971). Zircon can be found ubiquitously in sand-rich environments (e.g., van Es, 2008) – albeit in lower quantities than quartz and feldspar – and exhibits some unique features concerning luminescence-based dosimetry. Firstly, its internal U and Th contents are highly variable but typically amount to $10^2$–$10^3$ µg g-1 for Th and U (Zimmerman, 1971; Sutton and Zimmerman, 1976). If these internal U and Th levels far exceed the radioelement contents in the sedimentary matrix, the short-range (in particular α-

radiation) internal dose rate generated within the zircon dominates the total dose rate. External factors such as water content or spatial distribution of β- and γ-radiation that usually add additional uncertainty become far less important or negligible (Sutton and Zimmerman, 1976). The accrued dose in the zircons can be determined by comparing the natural TL with regenerated signals induced by α-irradiation in the laboratory (because of dominating internal α-dose rate; *classical approach*), where the dose rate is derived from analyses of the U and Th content of individual grains (Sutton and Zimmerman, 1976).

Despite the clear advantage of a dominant internal dose rate, the luminescence signal of zircons commonly suffers from anomalous fading, i.e., loss of signal over time in the high-temperature glow curve region that should be characterised by sufficient signal stability for dating (Wintle, 1973; Sutton and Zimmerman, 1976; Amin and Durrani, 1985; Templer, 1985b; Van Es et al., 2002a). Although Sutton and Zimmerman (1976) obtained TL ages of single zircon grains from ceramics largely in agreement with their known ages, they found all of the zircon ages were systematically underestimated, suggesting problems

with anomalous fading. Besides, radioelements in many natural zircon grains are not uniformly distributed but concentrated in layers resulting from zoning that occurred during initial crystal growth, where zones of high U abundance are anti-correlated



with TL sensitivity (Vaz and Senftle, 1971; Sutton and Zimmerman, 1976; Templer and Walton, 1985). This anti-correlation is difficult or even impossible to reproduce in the laboratory and leads to erroneously young ages. Furthermore, natural zircon grains vary greatly in their optical properties, and Van Es et al. (2002b) recommend using only transparent grains for TL dating

due to internal light absorption and radiation damage having substantially altered the dark and opaque grains. Especially for larger zircon grains, administering an external α-dose might result in underestimation of the natural dose and hence age, because internal absorption of luminescence is stronger in the case of natural signals originating from the entire grain as compared to signals stemming solely from the α-irradiated outer shell of the grain. Irrespective of zoning and optical absorption in zircon, luminescence models predict inaccuracies in dose determination by comparing natural TL/OSL signals with those

induced in the laboratory (Turkin et al., 2006). Appropriate heat treatments during or after irradiation might systematically improve inaccurate age estimates (Turkin et al., 2005; Turkin et al., 2006), but confirmation by follow-up studies is pending.

The *auto-regeneration* dating technique represents an alternative to the classical zircon dating approach (Sutton and Zimmerman, 1976). It overcomes most of the methodological difficulties outlined above (including anomalous fading and

zoning) and could circumvent the general limitations in age precision using quartz and feldspar. The basic principle of this technique, first applied to ceramics, is the zircon grains' capability to restore a measurable luminescence signal within months of storage in the dark (due to the high internal dose rate) after the natural signal's readout. In the linear dose-response range, the luminescence age can hence be calculated by (Templer, 1985a; Templer, 1986; Templer and Smith, 1988; Smith, 1988)

$$\text{Age} = \frac{\text{Natural luminescence}}{\text{Auto-regenerated luminescence}} \times \text{Storage time} \qquad (1)$$

The internal dose rate does not need to be quantified as a first approximation. With reference to Eq. (1), we note that the calculated final age is expected to vary based on whether the ages of individual grains are averaged or if the integrated luminescence of a set of grains (multi-grain aliquot) is utilised. This bias tends to yield a higher final age estimate when

averaging the ages of individual grains, especially in case of weak auto-regenerated signals (Ogliore et al., 2011). A few studies testified to the general applicability of the auto-regeneration method on ceramics with dating accuracies of TL ages of 4–45% (Templer, 1986), which may have the potential for significant improvements. For example, in the simplest scenario with a negligible external dose rate, the age uncertainty is only calculated from the counting statistics of two measured luminescence signals and their instrumental background levels (Templer, 1986), allowing dating precision of a few years. Notwithstanding

these initial achievements, significant obstacles in applying the auto-regeneration technique to zircons persist: (1) The routine separation of transparent zircon grains from bulk sediment in appropriate light conditions is challenging. Several procedures were proposed (Van Es, 2008; Sutton and Zimmerman, 1976; Smith, 1988) but they must be evaluated and thoroughly tested on various samples. (2) Despite some previous experimental work on the role of radio- and trace elements and associated defects in the generation of TL and OSL in zircon (Shinno, 1986; Godfrey-Smith et al., 1989; Laruhin et al., 2002), our





understanding of the relationship between the trace element composition of zircon and its properties relevant to auto-regeneration dating is still insufficient. (3) OSL ideally targets the signal with the best bleachability. Although some TL ages of detrital zircons, representing the last sunlight exposure event, are in reasonable agreement with independent age control (albeit not obtained with auto-regeneration, but using additive γ-irradiation; Van Es, 2008), to the best of our knowledge, the OSL emission of zircon has not yet been thoroughly tested or applied for dating purposes. (4) Templer (1986) and Smith

(1988) reported low TL and OSL signal levels after laboratory storage, precluding OSL from being used in auto-regeneration. Luminescence readers developed in the last two to three decades now allow for single-grain level stimulation at much higher power densities (>50 W cm$^{-2}$; Duller et al., 1999) than those used in previous studies (~0.002 W cm$^{-2}$; Smith, 1988), with the potential to substantially improve the signal-to-noise ratio for auto-regenerated zircon OSL.

To test the potential of the zircon auto-regeneration approach using state-of-the-art equipment, we address items (2), (3), and (4) listed above by characterising the TL and OSL signal of two zircon samples in terms of signal composition, rates of optical resetting as well as the ability to auto-regenerate. These data are supplemented by cathodoluminescence images and trace element analytics by laser ablation inductively coupled plasma mass spectrometry (LA-ICP-MS) to characterise the distribution and quantify contents of U and Th in zircons and to establish potential links between luminescence behaviour and

zircon geochemistry. All results are discussed regarding their implications for advancing the zircon auto-regeneration method towards a competitive, highly accurate and reasonably precise chronometric tool to investigate land surface changes and archaeological deposits on Mid to Late Pleistocene and Holocene timescales.

## 2 Materials and methods

### 2.1 Samples and sample preparation

We selected two zircon samples from different environments for the methodological investigations. Sample ZR229 was extracted from Fe-K syenogranite from the Mt. Blanc massif (France and Switzerland) that represents a short-lived magmatic pulse around 303 Ma (Bussy et al., 2000). Sample Can1 is detrital and originates from alluvial sand collected from Rio Carrao close to Canaima (Venezuela). It represents the weathering product of orthoquarzitic sandstone from the Pre-Cambrian Roraima Group, which forms the tepuis of the Gran Sabana. The sandstones from this group have a minimum age of 1.5–1.6

Ga (Briceño and Schubert, 1990) or ~1.8 Ga based on zircon U-Pb ages of mafic sills (Santos et al., 2003).

Zircons from sample ZR229 were extracted using heavy liquid and magnetic separation procedures detailed in Bussy et al. (2000), and no further grain size selection was applied. While the largest grains reach ~150 μm along their $c$-axis, most grains are considerably smaller (<80–100 μm). The sieved (63–100 μm) Can1 sample was treated with 10 % HCl to dissolve carbonates, run through a Frantz L-1 magnetic separator (slope 15°, tilt 15°, current through magnet 1.7 A; Porat, 2006), and

the non-magnetic fraction was then etched in 40 % HF at room temperature for 2 d, while the sample was continuously stirred. After 1 d, HF was replenished. After 2 d of HF etching, inspection of grains indicated the presence of light and dark-coloured





heavy minerals (potentially (fluor)apatite, fluorite or rutile), necessitating an additional HF etch for 3 d under the same conditions to optimise the zircon yield. This way, we could extract a few hundred zircons from an initial sample mass of ~500 g.

## 2.2 Mineral characterisation through cathodoluminescence and LA-ICP-MS

We prepared epoxy resin mounts for 42 and 43 zircon grains of samples ZR229 and Can1, respectively, by polishing them down to their equatorial plane using diamond paste (see Supplement, Figs. S2 and S3). The mounts were carbon coated, after which the polished grain surfaces were imaged using a CamScan MV 2300 SEM operated at an acceleration voltage of 10 kV, with a sample positioned at a working distance of ~40 mm. Panchromatic cathodoluminescence (CL) images were taken to visualise the growth structure of zircon grains (zoning) and to optimally select the location of the laser spots for LA-ICP-MS. The carbon coating was removed prior to LA-ICP-MS analyses.

LA-ICP-MS allows quantifying isotope ratios and elemental abundances along a cylindrical pit created through laser ablation into a solid surface. Here, we used a RESOlution SE 193 nm ArF excimer laser ablation system equipped with an S155 two-volume ablation cell (Applied Spectra, USA) and interfaced to a sensitive fast scanning triple quadrupole ICP mass spectrometer NexION 5000 (Perkin Elmer, USA). Laser repetition rate was set to 8 Hz and the on-sample energy density to 2.9 J cm$^{-2}$. The beam size was 24 µm for all analytical spots on zircon grains. The list of analysed isotopes included $^{27}$Al, $^{29}$Si, $^{42}$Ca, $^{49}$Ti, $^{85}$Rb, $^{88}$Sr, $^{89}$Y, $^{93}$Nb, $^{139}$La, $^{140}$Ce, $^{141}$Pr, $^{143}$Nd, $^{147}$Sm, $^{151}$Eu, $^{157}$Gd, $^{159}$Rb, $^{163}$Dy, $^{165}$Ho, $^{166}$Er, $^{169}$Tm, $^{172}$Yb, $^{175}$Lu, $^{178}$Hf, $^{181}$Ta, $^{201}$Hg, $^{204}$Pb, $^{206}$Pb, $^{207}$Pb, $^{232}$Th, and $^{238}$U. Their intensities were acquired sequentially using the mass spectrometer in the single quadrupole regime (Q1 as RF-only ion guide and Q3 as mass filter). Each measurement was reduced twice, to quantify the trace element composition and to estimate the $^{206}$Pb/$^{238}$U and $^{207}$Pb/$^{235}$U ages, respectively. The trace element data were reduced against the NIST SRM 612 soda-lime-silica glass standard (Jochum et al., 2011) as a primary standard for the determination of relative sensitivity factors. It was repetitively ablated at a beam size of 50 µm, using the same repetition rate and energy density as for the sample zircons. Quality control was ensured using the BCR-2G basaltic glass as a secondary standard. For internal standardisation, $^{29}$Si was selected. Its abundance was set up stoichiometrically (32.9 wt% SiO$_2$). The U/Pb ages were computed against the GJ-1 reference zircon (Boekhout et al., 2012; Ulianov et al., 2012) as a primary standard that was repetitively analysed at exactly the same ablation settings as for the sample zircons. As secondary standards for samples ZR229 and Can1, respectively, the Plešovice (Sláma et al., 2008) and Harvard 91500 (Wiedenbeck et al., 1995) reference zircons were used. The reference $^{206}$Pb/$^{238}$U ages of these two standards (337.13 ± 0.37 and 1062.4 ± 0.2 Ma at 2s confidence level) could be reproduced within 0.2 % ($n = 5$) and 2 % ($n = 7$), respectively. Both trace element and U/Pb data were reduced in LAMTRACE (Jackson, 2008; detection rules modified after Ulianov et al., 2016). The transient signals were carefully inspected for the presence of intensity spikes and domains that could contain inclusions, cracks, or other sources of chemical alteration, as well as for the consistency of the time-resolved U/Pb spectrum. As monitors of chemical alteration, we used $^{27}$Al, $^{85}$Rb, $^{42}$Ca and, in some cases, $^{139}$La. The canonical understanding of LA-ICP-MS usually implies rejection of altered



domains. In this study, however we created two parallel trace element datasets, based on the bulk signal (corresponding to the
integrated signal across the entire pit) or on specific signal segments selected so as to be devoid of alteration and contamination
as much as possible and thus representing the 'pure' zircon crystal structure. As the smallest entities relevant for luminescence
analyses are single grains, the bulk trace element abundances, of radioelements in particular, cannot be discarded *a priori*. For
Th and especially for U, the bulk and the structural values are, however, quite similar (Fig S1): high contents of these elements
usually present in the zircon crystal structure are difficult to change by alteration and contamination. Measured uncertainties
amount to ~2–5 % for values in the mg g$^{-1}$ range, i.e., for elemental abundances well above the critical level of detection. The
latter condition was met for most elements and analytical spots. For each zircon grain, either one or two analysis locations
(spots) were chosen, guided by inspection of individual grains' zoning as displayed in the CL images.

**2.3 Luminescence measurement parameters**

For single-grain OSL measurements, zircon grains were placed in the 300 μm diameter holes of Risø single-grain discs using
fine tweezers or a needle. Due to the small average size of the zircons, some holes contained more than one grain due to the
electrostatic charge of grains. This information was recorded to retrospectively relate luminescence characteristics to the
number of grains in individual holes. The TL signals from the holes were measured simultaneously and hence represent a
composite signal for the entire disc.

TL and OSL were recorded with Risø TL/OSL DA-20 readers equipped with a UV-sensitive EMI 9235QB photomultiplier
tube (PMT) showing a dark count rate of 35 ± 9 cts s$^{-1}$. TL signals were not filtered for auto-regeneration experiments. To
maximise the system's sensitivity and to also detect very dim TL signals close to the instrumental background level, the OSL
unit was replaced by the TL flange for these measurements, resulting in a smaller distance between the sample and PMT, and,
consequently, a larger solid angle captured by the detector. The heating rate was set to 1 K s$^{-1}$ in all TL experiments. TL spectra
were measured with an Andor Kymera 193i spectrograph coupled to a UV-enhanced Andor iXon Ultra 888 electron-
multiplying charge-coupled device (EMCCD) camera installed on the DASH of a Risø TL/OSL DA-20 reader. Further
technical details on the system are described in Bartz et al. (2023). All spectra reported are energy and efficiency calibrated.
The background relevant for TL measurements was recorded in a second run, using identical parameters as in the measurement
of the sample signal. OSL was stimulated by an Nd:YVO$_4$ solid-state laser emitting at 532 nm with a nominal power density
at the grain location of 45 W cm$^{-2}$ and recorded at 125 °C through a 7.5 mm Hoya U340 filter and with the PMT described
further above. The background relevant for OSL measurements was approximated by the signal recorded by the PMT before
the stimulation source was switched on, and no preheat was applied unless stated otherwise. TL and OSL signals were
regenerated in the laboratory by irradiation with a $^{90}$Sr/$^{90}$Y β-source characterised by a dose rate of 0.087 Gy s$^{-1}$ at the time of
measurement, as calibrated with Risø calibration quartz batch #200 (Hansen et al., 2015; Autzen et al., 2022). OSL signals
were decomposed using the function `fit_CWCurve()` of the R package 'Luminescence' (Kreutzer et al., 2012; Kreutzer,
2023; R Core Team, 2024).





# 3 Results

## 3.1 Cathodoluminescence

Representative CL images are shown in Fig. 1; a summary of all studied grains can be found in the Supplement. Most grains exhibit zoned growth, as indicated by mostly concentric areas of CL of various intensities. While concentric zones characterise
the zircons of sample ZR229, the growth structure of the Can1 zircons is much more complex, with initial (zoned) igneous or metamorphic zircon nuclei overgrown by younger, discordant domains (see Fig. 1(c) and Fig. S4). Considering the zircon typology scheme by Pupin (1980), it appears that both zircon samples differ in the alkalinity of the melt, but given the multiple growth and abrasional cycles preserved in the Can1 zircons, it remains unclear whether the more rounded shape of the latter relates to a lower crystallisation temperature, metamorphic overprinting and/or to physical abrasion (Markl, 2014). Overall,
CL allowed us to visualise the different origins of the two studied samples as indicated by contrasting grain morphology and zoning, reflecting the granitic origin of sample ZR229 on the one hand and the sedimentary transport of zircon grains from their source region in the Trans-Amazon Belt north of the Roraima basin and the diagenesis experienced by the detrital sample Can1 on the other hand (Santos et al., 2003).





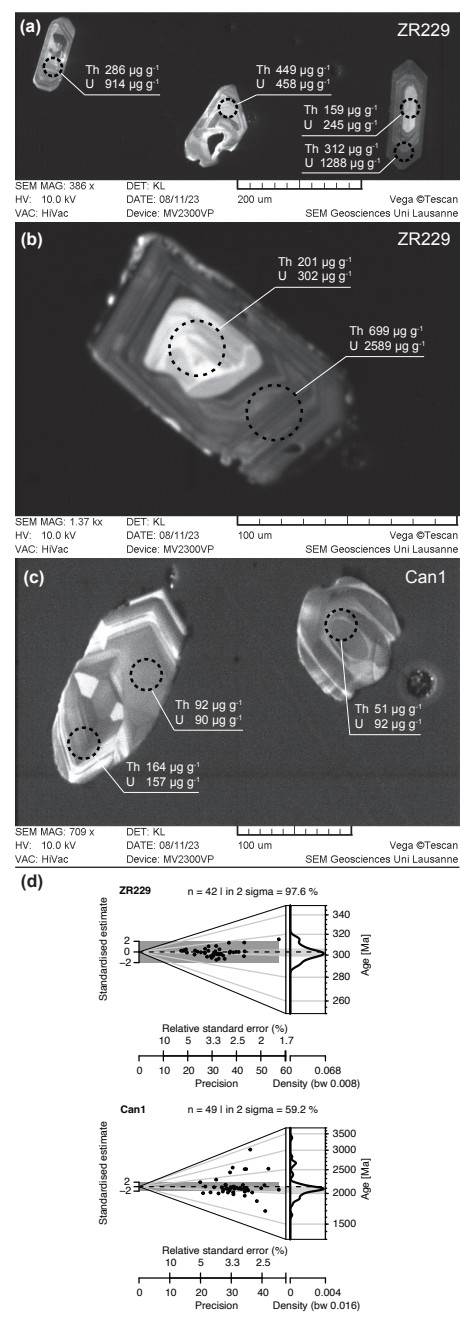

**Figure 1.** Cathodoluminescence (CL) images of selected zircons grains, with analytical spots from LA-ICP-MS and measured Th and U contents indicated. (a) and (b) Sample ZR229; (c) Sample Can1; (d) The Abanico plots (Dietze et al., 2016) show the U-Pb age distributions for the individual grains dated by LA-ICP-MS.



### 3.2 LA-ICP-MS

Analytical results confirmed that all 42 grains of sample ZR229 are zircons, while 4 out of the 43 grains of sample Can1 were discarded due to their elemental composition not aligning with the characteristic pattern for $ZrSiO_4$. Radioelement (Th, U) contents for the bulk material analysed in one spot (i.e., when integrating over the entire transient signal during ablation) are shown in Fig. 2 and given in detail in Tables S3 and S4. For sample ZR229, Th and U contents are positively correlated and vary by roughly two orders of magnitude (100–10,000 µg g$^{-1}$); about half of the analytical spots yielded values between 100

µg g$^{-1}$ and 1,000 µg g$^{-1}$ Th and U. The highest values recorded were 6990 µg g$^{-1}$ for Th and 8057 µg g$^{-1}$ for U, observed on two different grains, while the lowest contents measured for ZR229 were 97 µg g$^{-1}$ for Th and 226 µg g$^{-1}$ for U (also two different grains). The average abundances of Th and U for 48 analytical spots totalled 760 µg g$^{-1}$ and 2456 µg g$^{-1}$, respectively. Fig. 2 also suggests that there is no systematic difference of radioelement abundances between the core and the rim of zircon grains.

In contrast, Th and U contents are overall lower in zircon grains of sample Can1, with more than half of the grains giving contents between 10 µg g$^{-1}$ and 100 µg g$^{-1}$. The lowest and highest abundances for Th amount to 1 µg g$^{-1}$ and 404 µg g$^{-1}$ and for U to 15 µg g$^{-1}$ and 293 µg g$^{-1}$. Average values were 71 µg g$^{-1}$ for Th and 98 µg g$^{-1}$ for U, and the dispersion of data (Fig. 2) indicates that grain-to-grain variation in internal dose rate is smaller in sample Can1 compared to ZR229. Similar to sample ZR229, grouping the analytical spots into those located in the core of a zircon grain and those located in the rim resulting from

a later phase of zircon growth does not reveal any systematic differences in radioelement abundance between these zones (Fig. 2). However, particularly for sample Can1, the differentiation between the core and rim of zircon grains was often challenging due to frequent abrasion affecting significant portions of the grains. For both samples, the variation of radioelement contents within one grain (i.e., zonation) is smaller than the variation of averaged Th and U contents between grains. For instance, the average variation of the U content within grains of samples ZR229 and Can1 amounts to 75 % and 28 %, whereas the inter-

grain scatter of averaged U abundance adds up to 110 % and 52 %, respectively.



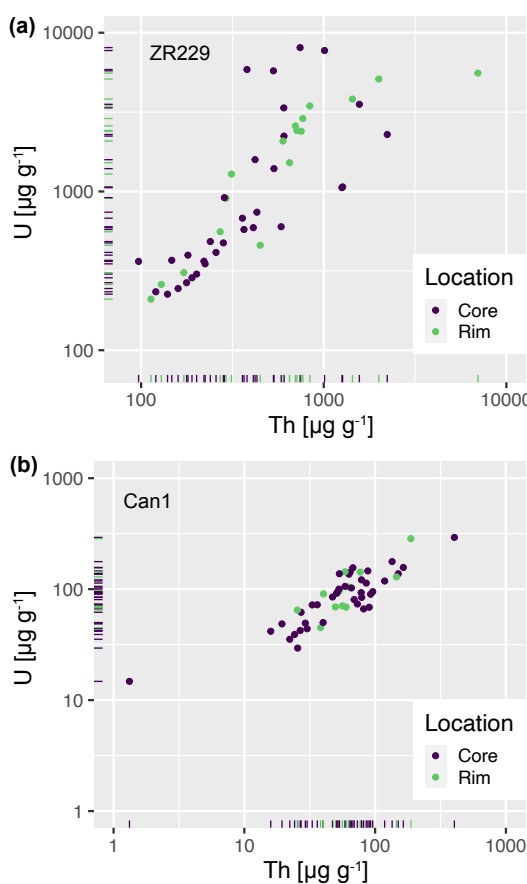

**Figure 2.** Radioelement contents (Th, U) for zircon samples ZR229 (a) and Can1 (b), as determined by LA-ICP-MS. Measured uncertainties are in the order of 2 % and not plotted here. Each dot represents one spot analysis located in the core or the rim of grains, while two spots were measured for some grains (in which case both core and rim were analysed).

Contrasting elemental abundances suggests that the structural Th and U contents in sample Can1 represent by far the most important part of the total, or bulk, Th and U budget of the individual zircon grains, whereas in sample ZR229 the structural and the bulk contents of these elements are less well matched (Fig. S1). There appears to be an indistinct trend in the data for both samples that hints at contaminated or altered (parts of the) grains being enriched in Th and U relative to their structural contents.

Bulk radioelement contents (as the smallest entities relevant for luminescence analyses are single grains) can be translated into internal dose rates that give rise to auto-regenerated luminescence. When using the conversion factors of Guérin et al. (2011), α-grain size attenuation factors by Brennan et al. (1991), β-grain size attenuation factors by Guérin et al. (2012), an assumed α-efficiency (*a*-value) of 0.03 ± 0.003 and a grain size range of 60–120 µm, average U and Th contents for samples ZR229




and Can1 result in internal α-dose rates of 181 Gy ka⁻¹ and 7.9 Gy ka⁻¹, respectively, as calculated with DRAC (v1.2; Durcan
et al., 2015). Internal β-dose rates amount to 34 Gy ka⁻¹ and 1.5 Gy ka⁻¹. Assuming a sedimentary matrix with representative
radioelement abundances of 1.5 wt.% for K, 10 µg g⁻¹ for Th and 3 µg g⁻¹ for U, the internal dose rate accounts for ~99 % and
~76 % of the total environmental dose rates for samples ZR229 and Can1, respectively.

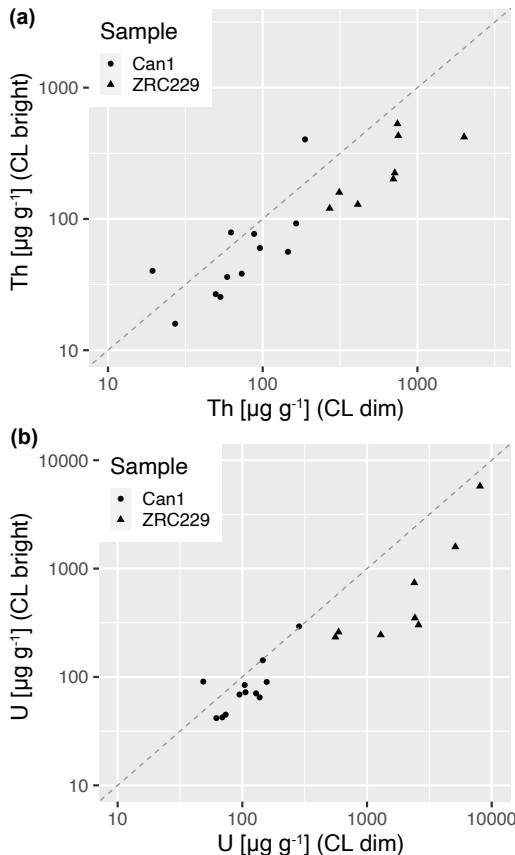

**Figure 3.** Th (a) and U (b) contents measured at two different analytical spots on one zircon grain. A relative comparison of
cathodoluminescence (CL) brightness for each grain allowed for analytical spots to be grouped into "CL dim" (plotted on the x-axis) and
"CL bright" (plotted on the y-axis); see Fig. 1 for example CL images. The dashed line represents the 1:1 line.

Previous studies found that the TL and CL sensitivity of domains in zircon grains characterised by high Th and U abundance
was lower than in domains with low radioactivity (Vaz and Senftle, 1971; Sutton and Zimmerman, 1976; Templer and Walton,
1985). This observation was explained by α-recoil damage of the crystal structure in zones of high activity. Here, in case
different domains within one zircon grain were analysed separately by two LA-ICP-MS spots, we grouped these two domains
into a dim and a bright zone (relative to each other), based on CL images. The determined radioelement contents for these
paired datasets are displayed in Fig. 3 and clearly support a correlation between CL intensity and Th and U abundance: Except



for a few grains, dim CL corresponds to high radioactivity and vice versa (see also Fig. 1). This trend is notably more pronounced for elevated radioelement abundances (corresponding to stronger radiation damage), while for domains

characterised by relatively low radioactivity factors other than Th and U contents may exert in an influence on CL intensity.

LA-ICP-MS analyses facilitated U-Pb dating of the zircon formation age for both samples, which – together with the radioelement abundance – might yield insights into potential metamictisation and its effects on luminescence properties. Seven out of 49 analytical spots of sample ZR229 did not yield concordant $^{206}Pb/^{238}U$ and $^{207}Pb/^{235}U$ ages (discarded from Fig. 1d). The main reasons for age discordance were the loss of radiogenic lead, presence of small amounts of common lead, or strong

heterogeneity within the ablated volume manifested by inconsistent Pb-U fractionation patterns and strong zoning including Th/U ratios <0.1 characteristic for metamorphic zircon grains (Yakymchuk et al., 2018) that yield a younger age than initial igneous crystallisation. For sample Can1, two out of 49 analytical spots were removed from the U-Pb dataset due to sampling of domains producing an age much younger than the average age (although their Th/U ratio did not indicate metamorphism). These grains are probably derived from younger source rocks and then mixed with older grains in the Roraima detritus. The

42 accepted measurements from sample ZR229 yield a weighted mean age of 302.4 Ma with an estimated uncertainty range between ± 1.1 Ma (measured) and ± 4.2 Ma (full range including systematic uncertainties) at 1s confidence level. This age estimate agrees well with the age of 303 ± 2 Ma for the Mont Blanc Granite given in von Raumer and Bussy (2004). Considering the full set of spots, we obtain a weighted mean age of 2,114 Ma for sample Can1with an estimated uncertainty ranging between ± 42 Ma (measured) and ± 50 Ma (full range including systematic uncertainties) at 1s confidence level. This

estimate is consistent with the minimum age of 1,600–1,500 Ma for the Roraima sandstone as obtained for dated diabase intrusions (Briceño and Schubert, 1990) and the U-Pb ages obtained for the Roraima Supergroup by Santos et al. (2003). Spot measurements yielding U-Pb ages different from those in the main age mode (Fig. 1 d) would allow further distinction of zircon sources, but for the sake of this study a bulk age of the sample is sufficient to constrain the time available to produce radiation damage. The age difference and the variance in Th and U contents between the two zircon samples studied here thus

cover roughly an order of magnitude, with the younger sample (ZR229) producing a much higher internal dose rate than the older sample (Can1).

### 3.3 TL spectra

Our initial experimentation involved applying various β-doses before TL spectrum acquisition, revealing that relatively high doses (>2,000 Gy) are required to generate a sufficiently intense signal suitable for spectral analyses. Figure 4 shows two

typical TL spectra of samples ZR229 (Fig. 4a) and Can1 (Fig. 4c) as contour plots, as well as the spectral composition of TL signals for two different temperature intervals. Both spectra exhibit a broad emission, spanning ~320–450 nm (centred at ~400 nm), and share an emission centred at ~640–650 nm. The former emission covers almost the entire recorded temperature range (25–400 °C) and displays an increase in intensity for glow temperatures >350 °C. In contrast, the latter attains its peak intensity within temperatures of 100–200 °C (ZR229) or 150–210 °C (Can1). In addition to these common features, the TL spectrum of



sample ZR229 yields narrow emission bands centred at 480 nm and 580 nm, which can be traced for glow temperatures
between 80 °C and 280 °C. This sample possibly shows an additional emission beyond 650 nm, as indicated by the broad
emission plateau in the wavelength region 620–700 nm (Fig. 4b).

**Figure 4.** TL spectra of samples ZR229 (a, b) and Can 1 (c, d) following a regenerative β-dose of 2,000 Gy and 3,200 Gy, shown as contour plots (a, c) and cross-sections for glow temperature intervals 100–175 °C and 300–350 °C (b, d). All data shown are background-subtracted as well as energy- and efficiency-calibrated.





### 3.4 OSL signal and its composition

We show typical OSL decay curves of samples ZR229 and Can1 in Fig. 5. The sensitivity (in cts s$^{-1}$ Gy$^{-1}$) and the optical decay
rate vary substantially from grain to grain, covering about two orders of magnitude. The background-subtracted, integrated
OSL signal over the entire stimulation period (0.25–2.75 s) is plotted against the number of grains in each hole of the single-
grain disc in Fig. 6. Comparing the signals obtained for holes without grains with those induced by one single grain suggests
that the fraction of grains that do not give rise to any significant OSL is higher in sample Can1 than in ZR229, in which case

all grains seem to produce measurable OSL following a β-dose of ~200 Gy. Furthermore, the counterintuitive observation that
measurements involving single grains yield, on average, larger OSL signals than the simultaneous measurement of several
grains can be best explained by the fact that placing just one grain in the hole of a single-grain disc was much easier to realise
for comparatively large zircons, while it was harder for smaller zircon grains that in many cases could not be separated from
each other easily due to electrostatic charging. Figure 6 hence implies that the OSL signal from large zircon grains tends to be

more intense than those from smaller grains and that there is a positive correlation between grain size/volume and OSL
produced. This trend is more pronounced for ZR229, while this sample also produces single-grain OSL signals with a median
more than four times larger than for sample Can1. This observation and the overall larger scatter in OSL intensities of sample
ZR229 between holes is consistent with the grain size spectrum of both samples, which is well constrained to 63–100 μm for
Can1, while no grain size separation was done for ZR229. The integrated OSL intensity along with the information on the

number of grains in each hole allows the average OSL sensitivity of one zircon grain for each of the two samples to be
estimated. It amounts to ~1.7 cts Gy$^{-1}$ for both samples for the summed OSL signal over the stimulation period of 2.5 s, with
estimated minimum standard deviations (due to simultaneous measurement of multiple grains in some holes) at 1$\sigma$ of 1.2 cts
Gy$^{-1}$. However, these averaged figures completely mask the observation that larger grains give rise to the strongest OSL signals
(see Fig. 6). Normalising OSL sensitivity to sample mass might be a better alternative.



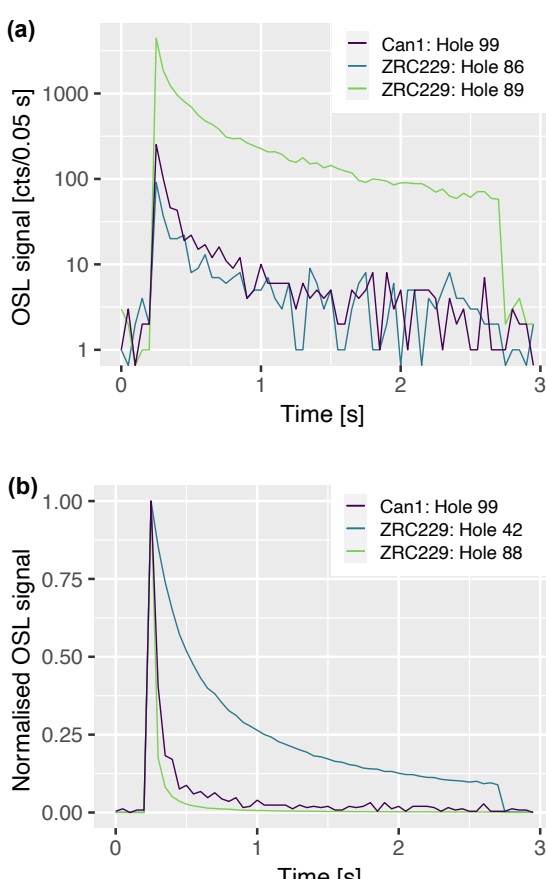


**Figure 5.** (a) OSL decay curves resulting from green laser stimulation of zircon grains in the holes of a single-grain disc following ~200 Gy (Can1) or ~540 Gy (ZR229) regenerative β-irradiation, demonstrating that OSL sensitivities of individual grains of one sample (ZR229) as well as of grains from different samples vary by almost two orders of magnitude. Some grains did not yield a signal above the background (not shown here). The first five channels (0–0.25 s) and the last five channels (2.75–3 s) represent the instrumental background recorded

before the stimulation light was switched on. (b) Normalised OSL decay curves of grains from both samples indicate that the OSL signal decays at very different rates on a grain-to-grain level.



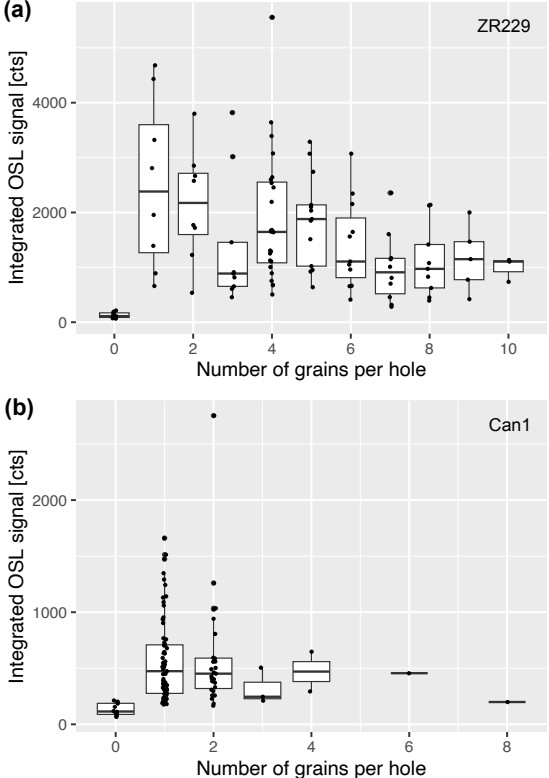

**Figure 6.** Integrated OSL signal (0.25–2.75 s) as a function of the number of grains per hole of a single-grain disc, as counted under an optical binocular, for sample ZR229 (a) and Can1 (b).

Previous studies suggested that the zircon OSL signal comprises several components characterised by different optical decay rates or photoionisation cross-sections (Smith, 1989), similar to later observations for quartz OSL (Bailey et al., 1997). To gain further insights into zircon OSL signal properties, we decomposed the regenerated (~200 Gy) continuous wave OSL decay curves obtained from measuring one or more grains in the holes of a single-grain disc. Optical decay rates were translated into photoionisation cross-sections, assuming the accuracy of the green laser's stimulation power density. About 25–34 % of OSL

decay curves could be fitted with one exponentially decaying component, while 59–65 % required two components (see examples shown in Fig. S7). Only 6–13 % of the decay curves necessitated three components for a satisfactory fit; exact numbers are provided in Table 1. The distribution of photoionisation cross-sections of the two 'fastest' components is displayed in Fig. 7 for samples ZR229 and Can1 as a histogram and probability density function (PDF). Calculated values for the photoionisation cross-section span a range from $2.4 \times 10^{-22}$ cm$^2$ to $3.2 \times 10^{-19}$ cm$^2$. Within uncertainty margins, the maxima in

the PDF of zircon photoionisation cross-sections are compatible with the s1 and s3 components of quartz OSL as defined by Singarayer and Bailey (2003). A fast component with photoionisation cross-sections in the order of $10^{-17}$ cm$^2$ (Singarayer and Bailey, 2003) could not be detected in any of the decay curves. It appears that heating of sample ZR229 to 180 °C prior to





OSL measurement does not substantially change the bimodality and values of photoionisation cross-sections (see Figs. 7a and 7b). Furthermore, whenever just one component could be fitted, its photoionisation cross-section is, in the great majority of
cases, smaller than $\sim 5 \times 10^{-20}$ cm$^2$, suggesting that for those grains, the fastest component (with a photoionisation cross-section in the order of $10^{-19}$ cm$^2$) is absent in the signal.

**Table 1.** Proportion of the components required to fit the OSL decay curves of the studied samples satisfactorily. The ZR229-PH sample was preheated to 180 °C prior to measurement.

| Sample | ZR229 [%] | ZR229-PH [%] | Can1 [%] |
| --- | --- | --- | --- |
| 1 component | 0.29 | 0.34 | 0.25 |
| 2 components | 0.65 | 0.59 | 0.62 |
| 3 components | 0.06 | 0.07 | 0.13 |




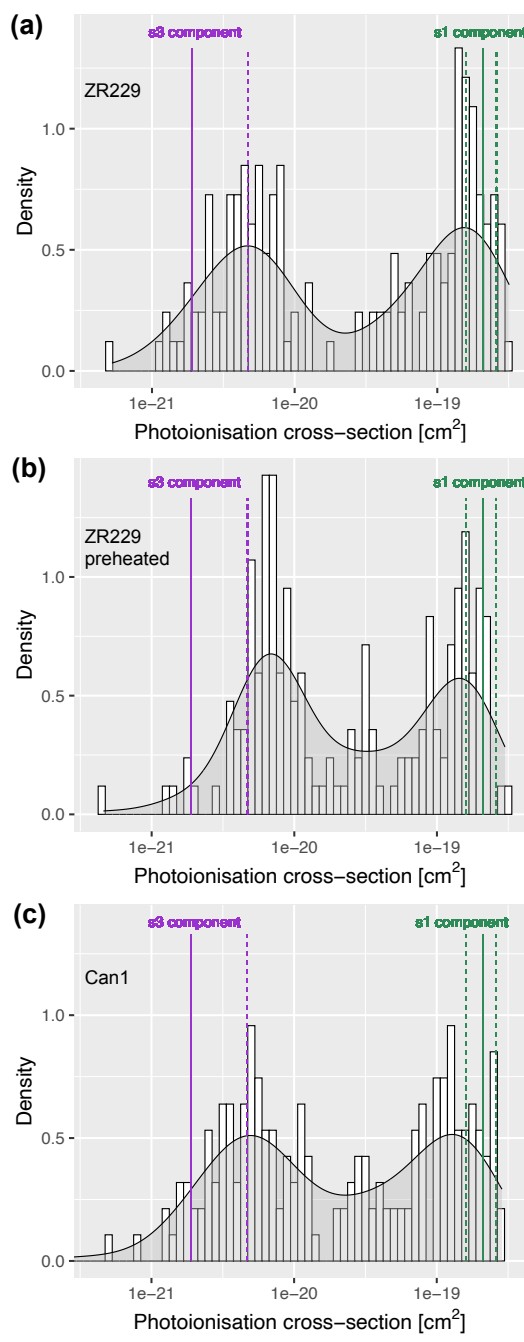

**Figure 7.** Distribution of photoionisation cross-sections obtained by fitting single exponential decays to OSL transients of samples ZR229 (a), ZR229 preheated to 180 °C (b) and sample Can1 (c). Green and purple lines indicate photoionisation cross-sections for the s1 component and the s3 component of quartz, as published by Singarayer and Bailey (2003), with dashed lines symbolizing the 1$\sigma$ confidence interval.



## 3.5 Bleaching of zircon OSL

The zircon OSL decay curve decomposition results suggest that the reduction rate of zircon OSL under sunlight exposure is slower than that of quartz OSL. To test this assumption and the suitability of the zircon OSL signal to date light exposure events, we conducted a bleaching experiment on sample ZR229. A single-grain disc populated with one or more grains per hole was repeatedly irradiated with a β-dose of ~220 Gy, exposed to the light from a Hönle UVACube400 (see emission spectrum in Fig. S8) and the residual OSL signal was measured. The light exposure times were set to 0, 10, 30, 60, 120 and 300 s. To distinguish OSL signal reduction due to fading from the reduction induced by light exposure, we ensured that the delay between irradiation and measurement was always kept at 16 min, irrespective of the bleaching duration. In this way, fading was identical in all bleaching cycles, and the net effect of bleaching could be isolated. Following the reading of the residual signal, a test dose cycle (administering a β-dose of 200 Gy) was inserted to correct for any sensitivity changes. This experiment was carried out once with sample ZR229 receiving a cutheat using an arbitrarily selected temperature of 180 °C directly following β-irradiation and once without preheating.

Experimental results include only positions in the single-grain disc yielding monotonic decay with bleaching time and giving integrated signals at 0 s bleaching time larger than 100 cts (~3 times the standard deviation of the background; Fig. 8). Although some scatter between residual signals of the same bleaching time can be observed, the boxplot in Fig. 8a confirms the reduction of the average residual signal by >50 % following 10 s light exposure. The zircon OSL signal decayed to negligible values (<3 % of the initial signal) after 300 s bleaching. Although the trend of OSL signal reduction with bleaching time is the same for the non-heated and the preheated samples, it appears that preheating removes part of the most light-sensitive OSL components.

Contrasting the bleaching characteristics of other commonly used luminescence signals for dating (Fig. 8b), zircon OSL bleaches at least one order of magnitude faster than any IRSL signal of feldspar but requires a few minutes more of sunlight exposure to be fully reset as compared to quartz OSL (Colarossi et al., 2015).




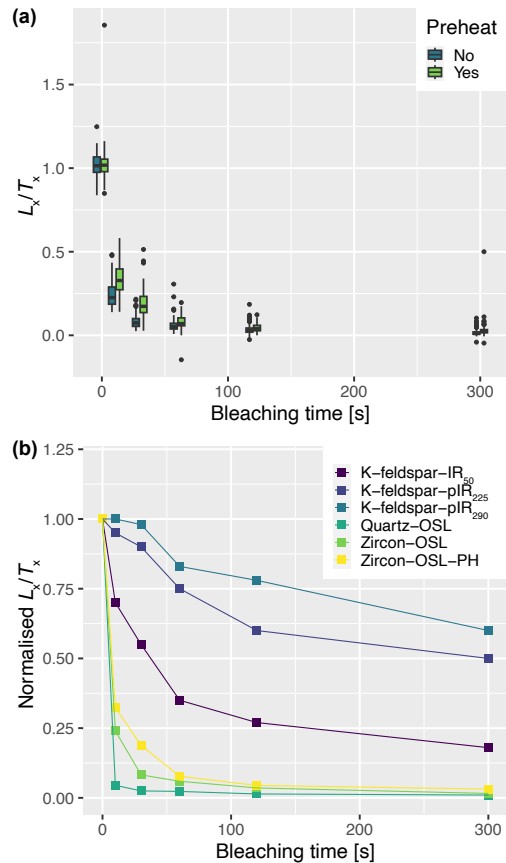

**Figure 8.** Results from the bleaching experiment with zircon grains of sample ZR229 placed in a single-grain disc. (a) Boxplots of all residual signals were measured following various bleaching times. (b) Comparison of zircon OSL residual signals as a function of bleaching time with those of other commonly used luminescence signals for dating. Zircon OSL data points represent the average of between 63 and 90 individual signals; the preheat temperature was 180 °C. The data for quartz OSL and K-feldspar IRSL signals were taken from Colarossi et al. (2015).

## 3.6 Auto-regeneration of zircon OSL and TL

Following the experiment on the optical resetting rate of zircon OSL, we investigate its signal's potential for auto-regeneration dating. One crucial requirement with this is the ability of zircon OSL to build up a measurable signal within a reasonable amount of time, ideally not exceeding 1–2 a. We prepared one single-grain disc for which all holes were filled with merely one zircon grain of sample ZR229 and a second one with each hole containing more than one grain of the same sample. All holes were initially stimulated for 4 s with the green laser (see Section 2.3) to erase inherited signals, and the two single-grain discs were then stored in the dark at room temperature (~20 °C). After storage periods of 64 d, 127 d, 291 d and 540 d, the auto-regenerated signal of the grains in holes 1–20, 21–40, 41–60 and 61–80 were measured, respectively. Although sample




ZR229 was prepared and handled under daylight conditions, some grains yielded an initial OSL signal significantly above
instrumental background, indicating that green laser stimulation is more effective in resetting the OSL signal than daylight. To
ensure that all zircon grains were set to zero before starting the auto-regeneration experiment and that we could also detect
weak induced OSL signals, we excluded holes from the analyses that showed an integrated OSL signal (0–2.5 s of the decay
curve) larger than 40 cts s⁻¹ during the initial screening of the bleached sample. The results of the OSL auto-regeneration

experiment are shown in Fig. 9, with the measured OSL signals for each hole represented by box plots. It is evident that over
a storage period of 540 d (~1.5 a) the OSL signal did not increase significantly but remained barely distinguishable from the
instrumental background.

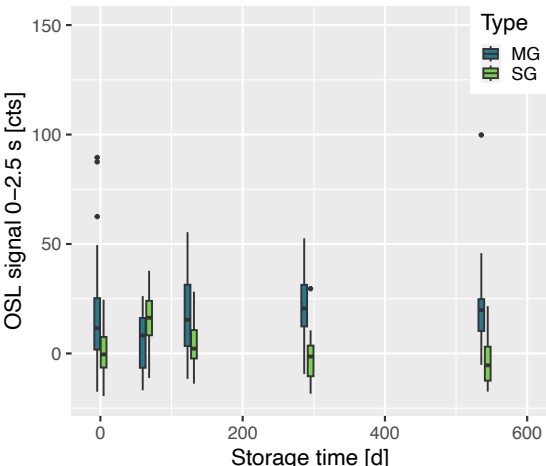

**Figure 9.** Boxplot of OSL auto-regeneration results for sample ZR229. Different colours represent analyses for single grains ("SG") and

multiple grains ("MG") loaded into the holes of a single-grain disc. A total of 20 holes in a single-grain disc were analysed for each storage
period.

Due to these dim auto-regenerated OSL signals and previously reported higher TL sensitivity of zircon, we investigated the
potential to exploit the TL auto-regeneration signal. To this end, one single-grain disc with one or multiple zircon grains of
samples ZR229 and Can1 was heated to 400 °C to erase all previously accrued doses and then stored in the dark for 21 d. The

auto-regenerated signals following this period are displayed in Fig 10. While sample Can1 shows a negative net signal for
glow temperatures <240 °C, indicating background reproducibility issues, sample ZR229 displays a notable positive TL signal
of up to 400–500 cts s⁻¹. It rises sharply for glow temperatures >100 °C and flattens out at 170–250 °C. The integrated net
signal from the temperature interval 0–280 °C is larger than $3\sigma$ of the corresponding background (which was measured four
times following the readout of the auto-regenerated signal), confirming that it is not an artefact caused by background

fluctuations. We repeated the TL auto-regeneration experiment for a storage period of 28 d with a very similar outcome for
both samples.




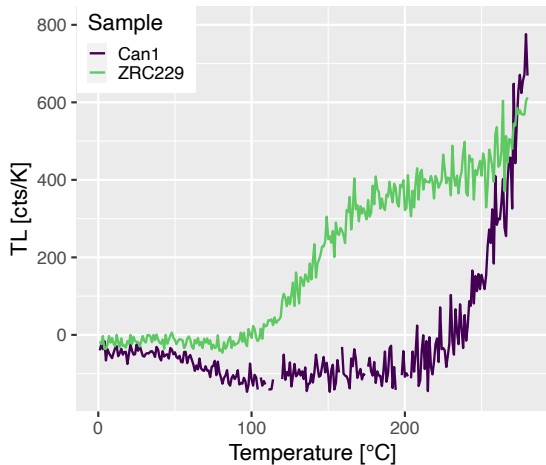

**Figure 10.** Auto-regenerated TL (not wavelength-filtered) of 505 zircon grains of sample ZR229 and 138 grains of sample Can1 following 21 d of storage without light exposure. Net TL signals are shown with their background (measured by a second glow) already subtracted.

**4 Discussion**

4.1 Geochemistry of zircons in relation to their luminescence

Amin and Durrani (1985) suggested that the TL spectrum of zircon changes as a function of radiation damage (metamictisation), with highly damaged zircons exhibiting only an orange-red emission, while less damaged zircons show an additional emission in the blue-green range (480 nm). This observation is consistent with spectrally resolved TL from the two
samples studied here. Sample Can1 is about one order of magnitude older than sample ZR229 and had more time to accumulate radiation damage. Indeed, this sample emits predominantly in the orange-red part of the spectrum, while ZR229 features additional emissions at shorter wavelengths. However, when the radioelement abundance in these two samples is taken into account, the situation becomes less clear. Averaged across all grains analysed by LA-ICP-MS, Th and U contents are higher by a factor of ~10 and ~20 for sample ZR229. This difference in radioelement abundance between the two samples should
have roughly equilibrated the integrated radiation damage effects and led to qualitatively more or less similar TL spectra. Therefore, other factors, such as the different geochemical composition of the samples, likely exert an additional influence on the TL properties (see Fig. 3 b). It should also be noted that the mineralogical history of the two samples is strikingly different, with the detrital zircons of the older sample Can1 exhibiting a much more complex pattern of grain evolution, possibly including several erosional and depositional cycles and associated moderate (but long-term) heating through diagenesis. These
divergent sample histories might have also left their imprint in the TL spectra of both samples, beyond the alteration induced by radiation damage. Vaz and Senftle (1971), Jain (1978), Amin et al. (1983) and Amin and Durrani (1985) found that high-temperature annealing above ~500 °C increases the luminescence sensitivity and may restore the TL spectrum as observed for less damaged zircons. Similarly, Godfrey-Smith et al. (1989) noted enhanced luminescence sensitivity following annealing,



accompanied by increased crystallinity, as detected by X-ray diffraction. Here, we annealed samples only to temperatures of
400 °C, in the course of which we could not detect any pronounced TL spectral and sensitivity changes. On the sub-grain
spatial scale, our data confirms the previous finding that both CL and TL intensity are negatively correlated with the
radioelement content in different domains of zircon crystals, especially for high U and Th abundances (Vaz and Senftle, 1971;
Amin et al., 1983; Templer and Walton, 1985; Nasdala et al., 2003).

The TL spectra of samples ZR229 and Can1 yielded a broad emission centred at ~400 nm across a wide range of glow
temperatures (room temperature to 400 °C) and a red emission at ~640 nm, peaking at 170–180 °C. In addition, sample ZR229
shows superimposed sharp emission lines at ~480 nm and ~580 nm that occur between 80 °C and 400 °C; these lines are not
present in the spectrum of sample Can1. Iacconi et al. (1980) explained the broad blue-violet emission observed in both our
samples as an intrinsic emission of the $SiO_4^{4-}$ ionic group in the $ZrSiO_4$ host lattice. An electron is detached from an oxygen
atom through irradiation and then trapped around a central Si atom. Recombination of the electron causes an emission at ~365–
380 nm (Iacconi et al., 1980; Godfrey-Smith et al., 1989), which is slightly shifted towards a shorter wavelength relative to
our observations. However, it is hard to identify a well-defined central wavelength due to the overall low intensity of this
emission in both samples. This emission occurs over a large range of temperatures and may be related to the observation and
theoretical prediction that the distance between oxygen and silicon atoms in $ZrSiO_4$ can vary, resulting in a (quasi-)continuous
distribution of activation energies. This effect was calculated and observed for the TL of synthetic zircon in the temperature
range of 100–350 K (-173–77ºC) (Iacconi et al., 1980). For the photoluminescence of zircon, Shinno (1986) explained broad
emissions in the spectrum occurring at various energies with centres created through α-decay of U and Th. The intensity of
such emissions was proposed to relate to the integrated radiation damage and hence to mineral formation age (or re-
crystallisation or annealing age), which in the case of the red TL emission at ~640 nm is consistent for the two samples studied
here. In comparison, such a relationship cannot be identified for the broad TL emission at ~400 nm. The narrow-band emissions
in the TL spectrum of sample ZR229 appearing at ~480 nm and ~580 nm are in agreement with many previously published
TL spectra of zircon (Jain, 1978; Kirsh and Townsend, 1987; Huntley et al., 1988; Van Es et al., 2002a) and generally attributed
to rare earth elements (REE) in the crystal lattice, with $Tb^{3+}$ (Jain, 1978), $Eu^{3+}$ (Nicholas, 1967)  or a combination of $Dy^{3+}$ and
$Tb^{3+}$ (Shinno, 1986; Laruhin et al., 2002) as the primary responsible activators. Laruhin et al. (2002) and Van Es et al. (2002a)
argue that $Dy^{3+}$ gives rise to narrow-band and intense TL emissions at lower glow temperatures (100–175 °C), while $Tb^{3+}$-
related emissions dominate the TL spectrum at higher temperatures (300–400 °C). Based on these previous observations, one
might expect a clear difference in Dy and Tb contents between samples ZR229 and Can1, given their different TL spectra
regarding the narrow-band emissions. Indeed, LA-ICP-MS results averaged across all analytical spots of a sample (see Tables
S3 and S4) demonstrate higher contents of both elements in ZR229 (Tb: 17 ± 12 µg g$^{-1}$, Dy: 212 ± 140 µg g$^{-1}$; uncertainty
range indicates the standard deviation at 1$\sigma$) as compared to Can1 (Tb: 6 ± 4 µg g$^{-1}$, Dy: 67 ± 44 µg g$^{-1}$). Although these results
potentially explain the absence of narrow-band emissions at 480 nm and 580 nm in sample Can1, they do not provide definitive
information on whether $Dy^{3+}$ or $Tb^{3+}$ accounts for those. By contrast, Godfrey-Smith et al. (1989) did not detect any clear





correlation between Dy (or any other REE) content and specific emissions and overall luminescence sensitivity. Given the geochemical and structural differences between the two studied zircon samples, it is interesting that they yield very similar OSL sensitivities and OSL signal composition (see Section 3.4.). As OSL was detected in the region ~270–380 nm, this could imply that both OSL and TL share the same recombination centres, i.e., those giving rise to the broad band centred around 400 nm in the TL spectra of both samples (Fig. 4). However, to address this question in detail, element analytical results of individual grains should be linked to their specific TL and OSL characteristics, which was not feasible in the scope of this study.

## 4.2 Auto-regeneration dating

### 4.2.1 Optical signal resetting

One of the main aims of this study was to test if the OSL signal of zircon is a promising candidate for auto-regeneration dating, given the technological advances made in measurement equipment and data analysis since the initial development of the method in the 1980s. An essential requirement for a mineral to successfully date a sunlight exposure event is the OSL signal's fast optical resetting. Our experiments demonstrate that for the sample studied, the initial portion of the zircon OSL signal is reduced to negligible levels after >300 s of exposure to simulated sunlight, in line with the bleaching rate of zircons in response to a 60 W tungsten filament lamp, resulting in 10 % residual signal after 120 s exposure (Smith, 1988). Future studies should investigate the bleaching rate for more samples, including coloured and opaque zircons. In terms of dating light exposure events of short duration or attenuated photon flux (e.g., slope wash, fluvial transport), this provides a distinct advantage over the TL signal of zircon, which was reported to show residuals of 35–65 % of the initial signal after 30 min exposure to a solar simulator (Smith, 1988) and over the infrared-stimulated luminescence signals of K-feldspar, which require much more prolonged light exposure to be fully bleached or that can be reduced only to an unbleachable residual (Chen et al., 2013; Kars et al., 2014). However, the comparatively rapid bleaching of zircon OSL starkly contrasts with the photoionisation cross-section values determined from laser-stimulated decay curves. Comparing published values for quartz with the values obtained in this study for zircon suggests the presence of signals with decay rates similar to the s1 and s3 components in quartz (Singarayer and Bailey, 2003). At least the s3 component would not reduce to zero following 300 s of daylight exposure. Most probably, this mismatch relates to the stimulation power density of the green laser used in our experiments, which might vary both over time and spatially, depending on the exact position of the grain in the hole of the single-grain disc, and that might deviate from its nominal value used to calculate the photoionisation cross-section. Consequently, a systematically wrong stimulation power density would lead to invalid photoionisation cross-section values.

### 4.2.2 Signal intensities

A second requirement for OSL auto-regeneration dating of zircons is a sufficiently high OSL sensitivity to induce well-quantifiable signals – i.e., signals significantly above the instrumental background – within a reasonable amount of time, say 1–2 a at maximum. In this regard, our results are somewhat disappointing, at least for sample ZR229, which exhibited no





noticeable OSL signal increase after 1.5 a of storage, corresponding to a self-dose of about 0.17 Gy based on the internal dose rate derived in Section 3.2. This finding applies to 'pure' single grains and multiple grains stored together and measured simultaneously. Furthermore, it is consistent with observations of the low OSL sensitivity of zircon samples Smith (1988) studied. Thus, an increase in optical stimulation power density of one to four orders of magnitude compared to experimental setups used previously (Smith, 1988; Smith, 1989) cannot improve the signal-to-background ratio sufficiently to render zircon OSL auto-regeneration practical. The low OSL signal yield might be explained by the fact that OSL selectively detects only a fraction of all emitted signals (those with energies higher than the stimulation energy), while TL allows for exploiting all emissions in the energy range for which the PMT is sensitive, thus maximising the TL yield. Smith (1989) argues that only a tiny portion of electrons evicted from their traps by light exposure recombine and produce OSL, while the majority of charge is re-trapped in shallow traps, giving rise to photo-transferred TL upon heating. Assuming an inherent similarity in TL and OSL emission spectra of zircons, our experimental data (Fig. 4) indicate that green laser stimulation, along with a detection window in the UV range, certainly captures only a small fraction of all photons emitted upon charge recombination. Moreover, for auto-regeneration measurements, care should be taken to physically separate individual zircon grains, e.g., in the holes of single-grain discs. The regenerated signals of multiple grains in a hole, as compared to 'pure' single grain measurements, are probably affected by cross-irradiation of grains during storage. Vaz and Senftle (1971) have shown that zones with high radioelement contents in zircon grains yield low TL signals and vice versa, an observation supported by our data (see Sections 3.1 and 3.2). On a grain-to-grain level, this anti-correlation would induce comparatively high luminescence signals in grains with low U and Th contents that are irradiated by neighbouring grains with higher activity, thus boosting the overall auto-regenerated luminescence output of a cluster of grains stored next to each other as compared to single zircon grains left for auto-regeneration in isolation from other grains. Although we found individual zircon grains are sometimes tricky to handle, future studies should thus follow the recommendation by Smith (1988) to store zircon grains separately for auto-regeneration.

### 4.2.3 Practical considerations

Given the difficulties of auto-regenerating zircon OSL signals within practical storage times, several strategies to further pursue this method are conceivable:

1. Apart from newly developed high-sensitivity luminescence readers, time-resolved OSL (TR-OSL) could improve the signal-to-noise ratio because it separates in time stimulation and detection of OSL and thus potentially enhances the signal-to-noise ratio (Chithambo, 2007; Bulur et al., 2014).

2. Increasing the OSL measurement temperature (e.g., to 150 °C) can potentially improve the signal-to-noise ratio through thermal eviction of electrons from shallow 'photo-transfer' traps (Smith, 1988). However, Smith (1989) noted that at higher temperature a harder-to-bleach OSL component is preferentially sampled, while, in contrast, at lower temperatures an easy-to-bleach component dominates the signal. This might have repercussions on dating poorly bleached samples.





3. It was shown that heating a zircon sample to temperatures >500 °C enhances its TL sensitivity by factors of 3–20 and OSL sensitivity by up to a factor of 200 (depending on heating conditions and sample properties), thereby annealing previously accumulated radiation damage and increasing crystallinity (Jain, 1978; Amin et al., 1983; Amin and Durrani, 1985; Godfrey-Smith et al., 1989). Especially geologically old zircons in a metamict state show low luminescence sensitivity that might, however, be (partially) restored by annealing. Therefore, Smith (1989) proposed a dating protocol in which the natural luminescence signal is measured first, followed by annealing the sample before auto-regeneration. The enhanced luminescence sensitivity is supposed to reduce auto-regeneration storage periods substantially, possibly to a few months for most zircon samples. This is compatible with findings by Templer and Smith (1988) and Smith (1989), who state that most heated zircons yield measurable auto-regenerated TL within one year, others even within periods as short as two months. One potential caveat is that heating changes the opacity of zircons, leading to apparent sensitivity changes through internal light absorption effects that, according to Templer (1985b) and Templer (1985a), amount to only <10 %, however. Further experiments on heat-induced sensitisation of zircon OSL are necessary to test the feasibility of this approach for optical resetting events.

4. Since we detected a significant auto-regenerated TL signal for one of the studied samples after three weeks of storage, combining zircon OSL and TL signals in an auto-regeneration approach might offer some pivotal advantages (Smith, 1988). Such an auto-regeneration protocol benefits from rapid bleaching of the natural zircon OSL, minimising the risk of age overestimation due to inherited signal contributions while shortening storage periods by using the more sensitive TL signal for auto-regeneration. An annealing step (e.g., at 500 °C) after reading the natural OSL erases any potential residual signals before storage and increases sample sensitivity. This changing luminescence sensitivity and using two different stimulation modes (OSL and TL) in the protocol necessitates introducing a test dose for normalising OSL and TL signals, respectively. However, zircon luminescence signals induced by laboratory irradiation suffer from anomalous fading (Wintle, 1973; Sutton and Zimmerman, 1976; Templer, 1985b) and exhibit unstable signal components so that Smith (1988) suggests a thermal pretreatment at 125 °C for 2 d to circumvent these problems and produce a laboratory signal most similar to the natural signal, as judged from the similarity of their TL glow curves.

5. As the efficiency of zircons to produce OSL (using the experimental parameters in this and most previous studies) is much smaller than their TL efficiency, photo-transferred TL (PTTL; Zimmerman, 1979) was proposed as a technique to measure small doses with greater efficiency and avoid anomalous fading (Bailiff et al., 1977) while at the same time benefitting from the better bleachability of the OSL signal (Smith, 1988). PTTL could be used in conjunction with the combined OSL/TL approach and allows for a much larger range of stimulation wavelengths, including those shorter than the 532 nm light used here, that are more efficient in completely erasing the OSL signal before regeneration.



### 4.2.4 Potential age range

Another important point to consider when assessing the practical avenues of zircon auto-regeneration dating is the covered age range. The estimated upper dating limit varies profoundly between studies. While Faïn et al. (1988) successfully dated the Puy de Dôme eruption (France) to $9.8 \pm 1.0$ ka and hence estimate that the method covers at least the last 10 ka (using additive laboratory irradiation), other authors argue for a much longer dating range. Smith et al. (1986) estimated the saturation dose of the 300 °C TL peak in zircon to ~4.5 kGy and inferred an upper dating limit of around 100 ka. Sutton and Zimmerman

(1976) measured linear α-dose response of zircon TL up to 20 kGy and saturation at ~100 kGy, whereas investigations by Vaz and Senftle (1971), Templer (1986) and Templer and Smith (1988) approximate the upper dating limit to ~400 ka. Based on the beginning of saturation of $Tb^{4+}$ centres at ~100 kGy, Laruhin et al. (2002) and Van Es (2008) calculated a dating range of 100 ka and beyond. For zircons, the great majority of the TL or OSL signal is induced by internal α-radiation. These comparatively heavy particles have a much lower efficiency in producing TL and OSL in quartz and feldspar than β- or γ-

radiation, resulting in an α-efficiency (*a*-value) of <1. Explicit α-efficiency values for zircon are not yet reported in the literature, except for the study by Van Es (2008), arguing that it is 1. Our own experiments (not reported here) and previous data on dose response in zircon, however, suggest an *a*-value of zircon of <1. Future studies should quantify zircon *a*-values and their variability between samples to estimate the range of linear α-dose-response from β- or γ-induced dose-response curves. Finally, considering that the auto-regeneration method only works reliably in the linear dose-response range, the actual

dating range is probably shorter than indicated by the saturation dose values in the studies cited above. Reconciling a linear α-dose range of 20 kGy (Sutton and Zimmerman, 1976) with the dose rates calculated from LA-ICP-MS data of the two samples studied here (see Section 3.2) gives an upper dating limit of between ~100 ka and ~2 Ma. The lower dating limit is defined by the signal-to-noise ratio obtained for the natural signal of young samples and, consequently, by the luminescence sensitivity. Due to residual zircon TL observed following sunlight exposure, only zircon OSL is suited for recent and subrecent sediment

samples thanks to its favourable bleachability (Smith, 1989). For heated zircons, the youngest age obtained with the auto-regeneration technique reported in the literature is $9.7 \pm 0.3$ a of a sand sample with an independent age of 9.8 a (Templer and Smith, 1988). Van Es et al. (2002a) claim that TL ages as young as 12 months can be achieved, although these authors use additive γ-irradiation to determine $D_e$ instead of auto-regeneration.

Lastly, although the internal dose rate of zircons constitutes the greatest part of the total dose rate, there might be scenarios in

which the external dose rate needs to be constrained for accurate age calculation. Such a case was encountered by Faïn et al. (1988) who dated the zircons contained in trachyte, where the external contribution accounted for 38 % of the total dose rate. Whenever zircons are extracted from sediments also containing quartz, the dose accrued in the quartz grains is identical to the external dose contribution in the zircon grains, provided that the infinite matrix assumption for β- and γ-radiation fields applies and that both quartz and zircons were reset entirely during the event to be dated.



## 5 Conclusions

Using zircon as a luminescence geochronometer offers distinct advantages over other minerals. Its high internal dose rate facilitates the use of auto-regenerated signals and significantly reduces the impact of the external dose rate and associated factors (e.g., gravimetric water content) on age. Here, we re-appraised the auto-regeneration approach, using up-to-date measurement technology, for its practical value by examining key features of the TL and OSL of two zircon samples with contrasting geochemical compositions. We summarise our findings as follows:

1. CL and LA-ICP-MS yielded highly contrasting radioelement contents between the two studied samples, spanning more than one order of magnitude. In most zircons, U and Th are non-uniformly distributed in zones, with zones low in radioactivity producing brighter luminescence and vice versa.

2. The TL spectra of the two samples show as common features a broad band at ~400 nm and a red emission at ~640–650 nm, while the geologically young sample with higher internal α-activity exhibits additional narrow-band emissions at ~480 nm and ~580 nm, usually attributed to rare earth ions acting as charge traps in zircon.

3. OSL sensitivity and decay rate vary substantially between grains of a zircon sample; however, both samples appear to share common signal components present in individual grains' signals in variable proportions.

4. Zircon OSL can be bleached by sunlight to negligible residuals within 300 s of exposure. Further studies, including opaque zircons, are pending.

5. Whereas a storage period of 1.5 a was not long enough to auto-regenerate a zircon OSL signal (for the high-activity sample), a measurable auto-regenerated TL signal could be obtained within 21 d. Special emphasis needs to be placed on accurate determination of background levels and their reproducibility.

6. A protocol combining the natural OSL signal with auto-regenerated TL, following Smith (1988), appears to be the most promising one in rendering the methodology practical and fostering a more widespread use in Quaternary geochronology.

**Code and data availability.** All raw and partially processed data and R codes are available at doi.org/10.48657/705a-g067.

**Author Contributions. CS** Conceived the study, carried out OSL, TL and auto-regeneration experiments, analysed and visualised the data, acquired funding, provided supervision to TH, prepared the first manuscript draft and organised the writing process with contributions from all authors, **TH** Carried out the bleaching experiments as part of his bachelor thesis, **PH** Developed zircon purification procedures, validated the data, and contributed to writing and editing, **AU** Planned and carried out LA-ICP-MS analyses and helped in data processing and interpretation, **BP** Assisted with re-fining zircon separation procedures and contributed to editing, **GK** provided supervision to TH, validated the data and contributed to writing and editing, **SK** Validated the data, acquired funding and contributed to writing and editing.



**Competing interests.** GK is a member of the Editorial Board of Geochronology. The authors also have no other competing interests to declare.

**Financial support.** This work was supported by the Swiss National Science Foundation (SNSF grant number
200021E_209643 / 1) and the German Research Foundation (DFG project number #505819035). SK received additional funding through the DFG Heisenberg programme # 505822867.

**Acknowledgements.** We would like to thank Thomas Grocolas, Dr Goran Andjic, Prof Othmar Müntener and Prof François Bussy (all University of Lausanne) for assistance with preparing the zircon mount, cathodoluminescence analyses, uncomplicated access to the LA-ICP-MS laboratory and providing sample ZR229, respectively. Additionally, we wish to
acknowledge the fruitful discussions with Prof Ludwig Zöller (University of Bayreuth) and his initial idea of re-addressing the zircon auto-regeneration method.



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
