# Peer review of "Zircon luminescence dating revisited"

_Geochronology, 2024_

## Author Comment (AC1)

**Reply to the comments by S. Tsukamoto on the manuscript gchron-2024-10**

First, we would like to thank you for your thoughtful comments on our manuscript. Before individually replying to each point further below, we would like to take the opportunity to elaborate that the present manuscript serves a two-fold purpose. First, it is meant to re-view and re-introduce the zircon auto-regeneration method, a dating technique proposed decades ago but abandoned since. Second, we present new data on zircon TL and OSL in light of the auto-regeneration method and new technical achievements that help assess the methodology's advantages, disadvantages and practical aspects. Our reply to the referee's comments should be seen in the light of this manuscript structure.

- ▪ Number of grains required for successful OSL auto-regeneration and storage time required to achieve an age precision of a few years

Concerning the number of grains and the storage time necessary to successfully auto-regenerate zircon OSL signals within a reasonable time (say 1-2 years), we add an explanatory note in the final version of the manuscript. Our further response to this point is divided into one section on cross-irradiation effects and practical aspects of measurement and one section on considerations regarding the impact of grain number and storage time on age precision.

1. It is worth noting that increasing the number of zircon grains in the hole of a single-grain disc may lead to higher bulk signals of regenerated OSL from this hole. However, due to reasons explained in the manuscript in section 4.2.2, lines 510-518 (cross-irradiation of grains of different OSL sensitivity) the measured signal cannot be directly compared to the zircon's natural signal (that was most likely accrued without the effect of cross-irradiation by neighbouring grains). This would lead to over-estimated regenerated OSL signals and age underestimation. This cross-irradiation effect makes using the combined signal of multiple grains stored in one hole of a single-grain disc difficult. To avoid cross-irradiation, zircon grains would have to be physically separated for auto-regeneration storage and then measured simultaneously, which could be achieved by conventional LED stimulation in luminescence readers (i.e., not using the single-grain stimulation mode), with the drawback of a much lower stimu-lation power density and decreased signal-to-noise ratio. Using commercially available luminescence readers, a maximum of 100 zircon grains could be measured at once in this way. Given that we could not discriminate regenerated OSL signals from the instrumental background for any of the 80 individual grains stimulated with a power density about three orders of magnitude higher than possible with LEDs, whether the integrated light sum of 100 grains would be measurable remains questionable. We emphasize, however, that the situation might be different for other zircon samples, and clearly more data is required regarding sample-to-sample variability in OSL regeneration behaviour.

2. In addition, and beyond the initial experiments presented in this manuscript, some further general considerations on the role of the grain number, the storage period, the background, the level at which a signal is detected and the way the age is computed seem worthwhile.

   a. For a given population of zircons (for a given zircon sample), assuming ordinary Poisson distribution for all gross signal and background intensities, in the framework of pure paired measurements (gross signal and background of equal duration for each of the analysed grains), we elaborate the following formalism:

   for a particular zircon:
   $$t = t_s \left( \frac{N - B}{N_s - B_s} \right)$$
   where:

$t$ = age of optical / thermal event to be estimated
$t_s$ = storage, or regeneration, time after which stimulated signal is extracted
$N$ = gross signal intensity for age
$B$ = background intensity paired to the acquisition of $N$
$N_s$ = gross signal intensity upon stimulation
$B_s$ = background intensity paired to the acquisition of $N_s$

for a population (sample):

$$t = t_s \left( \frac{\overline{N} - \overline{B}}{\overline{N_s} - \overline{B_s}} \right)$$

Where all gross signal and background values represent arithmetic means of the corresponding values for the individual zircon grains (as those were measured).

Consequently, for the standard deviation $s(t)$ of the age for the whole sample, we have:

$$s(t) = t_s \, s \left( \frac{\overline{N} - \overline{B}}{\overline{N_s} - \overline{B_s}} \right)$$

For the right term of this equation, a trivial derivation results in the following formalism:

$$s \left( \frac{\overline{N} - \overline{B}}{\overline{N_s} - \overline{B_s}} \right) = \frac{1}{\sqrt{k}} \left( \frac{\overline{N} - \overline{B}}{\overline{N_s} - \overline{B_s}} \right) \left( \frac{\overline{N} + \overline{B}}{\left(\overline{N} - \overline{B}\right)^2} + \frac{\overline{N_s} + \overline{B_s}}{\left(\overline{N_s} - \overline{B_s}\right)^2} \right)^{1/2}$$

where $k$ is the number of grains.

Obviously, all mean intensity values do not depend on the number of grains.

Consequently, $s(t)$ is inversely proportional to the square root of the number of grains. For example, to improve (i.e., to decrease) $s(t)$ twice, we need to increase the number of grains fourfold.

The relationships become somewhat less straightforward, if we increase $t_s$. How will the product

$$t_s \, s \left( \frac{\overline{N} - \overline{B}}{\overline{N_s} - \overline{B_s}} \right)$$

then respond?

Increasing $t_s$ proportionally and linearly increases $\overline{N_s}$, while, approximately, the term

$$s \left( \frac{\overline{N} - \overline{B}}{\overline{N_s} - \overline{B_s}} \right)$$

is inversely proportional to the square of $\overline{N_s}$. This proportionality can be further emphasized, if we simplify the full formula:

$$s\left(\frac{\overline{N}-\overline{B}}{\overline{N}_s-\overline{B}_s}\right) \approx \frac{1}{\sqrt{k}}\left(\frac{\overline{N}-\overline{B}}{\overline{N}_s-\overline{B}_s}\right)\left(\frac{\overline{N}_s+\overline{B}_s}{\left(\overline{N}_s-\overline{B}_s\right)^2}\right)^{1/2} = \frac{1}{\sqrt{k}}\frac{\left(\overline{N}-\overline{B}\right)\left(\overline{N}_s+\overline{B}_s\right)^{1/2}}{\left(\overline{N}_s-\overline{B}_s\right)^2}$$

Consequently, as an approximation, we can conclude that $s(t)$ is inversely proportional to the storage time. This relationship becomes especially well defined for low-background values: increasing the storage time twice decreases $s(t)$ twice. The exact relationship is still given by the full formula preceding this paragraph.

b. Concerning the measurement of the background and the discrimination of an auto-regenerated signal, it may appear advantageous to apply the definition of the critical value of detection $L_C$ ("the net signal level above which an observed signal may be reliably recognised as "detected""; Currie, 1968, as well as the original literature on tests for the equality of two Poisson means) and the determination limit $L_Q$ ("the level at which the measurement precision will be satisfactory for quantitative determination"; Currie, 1968). These measures provide guidance on whether an auto-regenerated signal was detected and if it can be quantified to calculate an age. While this situation applies to single measurements, it is important to state that non-detection does not mean a lack of signal from the sample, it only means that the probability to obtain a particular gross signal intensity as a fluctuation of the background intensity is higher than a particular threshold. Those non-detected signals could still be averaged across many measurements to compute an auto-regenerated signal. This question is often re-iterated in the literature of metrology, as far as the computation of mean values from weak individual signals is concerned (e.g., Currie, 1995, 1999).

c. The problematics of weak signals, such as our regenerated OSL signals, also raises questions regarding the way we average our data. There are generally two ways of computing the age from auto-regeneration measurements on $k$ single zircon grains. For natural signal intensities $N_i$ and auto-regenerated signal intensities $N_{s\,i}$ obtained after storage time $t_s$, and for the paired background intensities $B_i$ and $B_{s\,i}$, respectively, we may use either of two approaches, the mean-of-ratios or the ratio-of-means approach.

Mean-of-ratios:   $t_{\text{mean}} = t_s \dfrac{\sum_{i=1}^{k} \frac{N_i - B_i}{N_{s\,i} - B_{s\,i}}}{k}$

This formula applies to averaged single-grain signals, while the following formula refers to signal summation of all natural and regenerated grains before computing the age.

Ratio-of-means:   $t_{\text{mean}} = t_s \dfrac{\sum_{i=1}^{k} N_i - B_i}{\sum_{i=1}^{k} N_{s\,i} - B_{s\,i}}$

The mean-of-ratios estimator yields systematically higher values than the ratio-of-means estimator (e.g., Ogliore et al., 2011):

$$E\left[\frac{X}{Y}\right] \geq \frac{E[X]}{E[Y]}$$

The stronger the signal fluctuations between the different grains are, the stronger is the above inequality. Overall, it is not yet quite clear if the difference might be significant, given the comparatively large uncertainty of the TL or OSL auto-regeneration method. For hardly detectable regeneration signals and signals below detection, it should be significant, though, resulting in the appearance of a bias relative to the accurate age, and also strongly increasing

the age uncertainty due to the division by near-zero values in the mean-of-ratios approach. While for well detectable regeneration signals (large positive denominator: the hyperbolic function is easily linearised) the uncertainties of the mean-of-ratios and ratio-of-means estimators coincide, which is an ordinary exercise from the field of uncertainty propagation based on partial derivatives. To conclude, in the future research, it may be worth paying attention to how exactly we derive the age from auto-regeneration measurements. The relevant discussion, however, extends well beyond the scope of the present manuscript.

- Anti-correlation of U content and TL sensitivity as reason for age underestimation

Thank you for this comment, showing us that the manuscript would benefit from more explanation. In the case of zoning, some domains of a zircon crystal are characterised by higher U contents than other parts, while the U-rich domains yield a lower TL signal (despite the same age) than the U-poor domains. This means that the TL signal from the U-poor zones of the grain (with a low dose rate) dominates the measured bulk signal and, hence, the determined natural dose. In the 'conventional' TL dating of zircons, the internal dose rate of grains is estimated via their U and Th content, not taking into account that most of the measured TL signal is correlated with the U-poor zones and not with the average radioelement content. Thus, by calculating the age, one divides by a dose rate that is too high and not representative of the U-poor zones, giving most of the light and ultimately resulting in age underestimation (Sutton and Zimmerman, 1976). However, the dose for the low-U domains is not entirely defined by their low U abundance, as there might be a radiation flux from the neighbouring high-U domains.

- Sample preparation procedure

In the revised version of the manuscript, we provide details on zircon extraction from sample ZR229.

- Terminology: 'structural' vs. 'crystal'

In the revised version, we homogenise our terminology and use 'structural' throughout.

- Fig. 3: Trend in CL intensity vs. U and Th abundance

As CL intensity is estimated only qualitatively, the trend in Fig. 3 refers to differences in U and Th abundance between the two analysed domains of a zircon grain. In the revised version of the manuscript, we rephrased the respective sentence to avoid confusion.

- Explanation of one large zircon grain giving more OSL than several smaller ones

Thank you for this comment. Also, it was initially puzzling for us, and one potential explanation could be that – at least for two studied samples – the luminescence yield is proportional to grain volume. Another, less obvious reason could be that crystal growth conditions for large grains favoured the incorporation of luminescence-relevant impurities or the creation of defects. As the information that large grains tend to give brighter luminescence signals than a couple of smaller grains might be of relevance for practitioners, we would prefer to keep Fig. 6 in the manuscript. We add a note in the discussion of the revised manuscript to link Fig. 6 to the discussion.

- OSL components

Thank you for this comment. We are aware that signal component decomposition is a mathematical process that does not necessarily lead to physically meaningful results. However, given the silicate nature of both zircon and quartz, a comparison of photoionisation cross-sections of these two minerals seems worthwhile. Nevertheless, we understand that caution is required when directly relating the optical signal decay rates of different OSL components of the two minerals and adapt the text of the manuscript accordingly.

- Black dots in Fig. 9

The black dots in Fig. 9 represent outlier values, i.e., those that are beyond 1.5 x the interquartile range (Q1 – 1.5 x interquartile range or Q3 + 1.5 x interquartile range). This is a common way of plotting box plots

- Useful stimulation and detection wavelengths for time-resolved zircon OSL measurements

Since stimulation and detection wavelength may overlap when using time-resolved OSL, it seems plausible to focus on the most intense OSL emission of zircon. Given that probably the same recombination centers are involved in production of TL, OSL and PTTL (Kristianpoller et al., 2006) and most intense emissions were recorded between 350 nm and 500 nm and at ~620 nm for the two samples studied here, these two detection regions seem plausible for TR-OSL (while the latter would correspond to Stokes-shifted luminescence). Regarding the stimulation wavelength, previous studies (Bulur et al., 2014) used 445 nm laser stimulation, and most likely slightly shorter (see Kristianpoller et al., 2006) and longer wavelengths (e.g., the 470 nm LEDs in commercial luminescence readers or also a Nd:YAG laser at 532 nm or 266 nm) appear suitable choices for TR-OSL measurements as well.

- Discussion on the zircon luminescence dating range

We understand that aspects of the dating range were not part of the experimental data presented in this manuscript. However, given the scope of our manuscript and as outlined in our initial comment, we consider this manuscript also as an effort to 're-introduce' this somewhat forgotten methodology and make it more popular in the community.  As such, we think that readers might appreciate a more comprehensive overview of the method, without consulting additional literature. The expected age range of the method is an important key feature, as experienced during multiple conversations with peers. Therefore, although section 4.2.4 does not directly refer to experimental data presented in the study, it might provide the audience with valuable practical considerations about the method's potential and whether or not it is worthwhile to consider it for own use.

- Typos

In the updated version of our manuscript, we correct all typos identified by the reviewer.

References

Bulur, E., Kartal, E., Saraç, B.E., 2014. Time-resolved OSL of natural zircon: A preliminary study. Radiation Measurements 60, 46-52.

Currie, L.A., 1968. Limits for qualitative detection and quantitative determination. Application to radiochemistry. Analytical Chemistry 40, 586-593.

Currie, L.A., 1995. Nomenclature in evaluation of analytical methods including detection and quantification capabilities (IUPAC recommendations 1995). Pure Appl. Chem. 67, 1699-1723.

Currie, L.A., 1999. Detection and quantification limits: origins and historical overview, Anal. Chim. Acta 391, 127-134.

Kristianpoller, N., Weiss, D., Chen, R., 2006. Effects of photostimulation in natural zircon. Radiation Measurements 41, 961-966.

Ogliore R.C., Huss, G.R., Nagashima, K., 2011. Ratio estimation in SIMS analysis. Nuclear Instruments and Methods in Physics Research B 269, 1910-1918.

---

## Author Comment (AC2)

**Reply to the comments by J. Durcan on the manuscript gchron-2024-10**

We wish to thank the referee for the careful review of our manuscript and respond below to the points raised.

- Add a sentence to explain why there is a negative relationship between U concentration and TL sensitivity

The negative relationship between TL sensitivity and U content is thought to arise from damaging the crystal structure as a result from alpha-recoil, a process that can be reversed by high-temperature (950 °C) annealing (Vaz and Senftle, 1971; Amin et al., 1983). This information is added in the introduction of the revised manuscript. The same mechanism applies to radiation damage caused by internal Th, although the overall effect will be smaller than that induced by U, due to commonly lower Th abundances in zircon (compared to U) and the relatively smaller number of alpha-decays in the Th decay chain (about 25 % of the U alpha-activity if normalised to unit content).

In more general terms, the TL produced in a zircon following a storage period $t_s$ may be written as

$$\mathrm{TL} \sim c_U \cdot N_{TL} \cdot t_s$$

where $c_U$ is the U content and $N_{TL}$ the number of defects responsible for TL generation that are currently present in the crystal.

For simplicity, let us fix $t_s = const$ and study how do TL and $\mathrm{TL}_{sens}$ depend on $c_U$, provided all our experiments are conducted for the same storage time:

$$\mathrm{TL} \sim c_U \cdot N_{TL}$$

For a given zircon crystallisation age, we can relate $c_U$ to the number of U-induced alpha-decays $N_{dec}$ accumulated in the crystal since the time of its crystallisation:

$$\mathrm{TL} \sim k \cdot N_{dec} \cdot N_{TL}$$

where $k$ is a coefficient of linear proportionality. However, the number of currently available TL-relevant defects $N_{TL}$ is also a function of the number of decays (as the latter cause a destruction of TL-relevant defects):

$$\mathrm{TL} \sim k \cdot N_{dec}(N_{TL\ \text{initially present at the time of crystallisation}} - \varphi N_{dec})$$

where $\varphi$ is a parameter signifying the mean number of destroyed TL-relevant defects per alpha-decay. As a first approximation, $\varphi$ could be considered constant through geological time, though this only holds true provided the alpha-tracks do not interact with each other. For high U contents and old crystallisation ages, the tracks can intersect or otherwise interact with each other, thus reducing the mean number of destroyed TL-relevant defects per alpha-decay while the zircon is aging. The relevant modelling includes recurrent relationships that are easy to program as a computer code (numerical integration), but less easy to present as an analytical solution (i.e., as a single and compact formula integrating the number of the destroyed TL-relevant defects over the geological age). In this reply, for the sake of simplicity, we will consider $\varphi$ as a constant, but we understand that the exact representation of it is worth a deeper study.

The above equation is of the form $y = ax - bx^2$, i.e., TL should first increase with $N_{dec}$ before reaching a maximum at $a/2b$ and then decrease.

Let us now consider the TL sensitivity $TL_{sens} = TL/dose$. We have:

$$TL_{sens} \sim \frac{k \cdot N_{dec}(N_{TL\ initially\ present\ at\ the\ time\ of\ crystallisation} - \varphi N_{dec})}{dose}$$

For a given storage time, the dose is linearly proportional to $c_U$, as is the number of decays $N_{dec}$ that were accumulated in the crystal since the time of crystallisation:

$$TL_{sens} \sim \frac{c_U(N_{TL\ initially\ present\ at\ the\ time\ of\ crystallisation} - \varphi N_{dec})}{d \cdot c_U}$$

where $d$ is, again, a coefficient characterising the linear proportionality between the U content and the dose during the storage period.

The final relationship can be given as follows:

$$TL_{sens} \sim \frac{N_{TL\ initially\ present\ at\ the\ time\ of\ crystallisation} - \varphi N_{dec}}{d}$$

or simply

$$TL_{sens} \sim N_{TL\ initially\ present\ at\ the\ time\ of\ crystallisation} - \varphi N_{dec}$$

The above relationships provide a framework for further discussion, but do contain simplifications that are not limited to the exact conduct of the parameter $\varphi$ that we assumed to be constant. Another level of detail comes into play once we consider the statistical performance of $\varphi$ and possibly other parameters involved in the above calculus, which are in fact controlled by ordinary Poisson processes over space or over time.

- HF etching of zircon grains

The procedure of HF etching of heavy mineral samples to isolate zircons has been described in the literature, though not in great detail (e.g., Sutton and Zimmerman, 1976; Smith, 1988; Templer and Smith, 1988). While previous studies have typically used HF etching for 1 h, we found that, particularly for sample Can1, longer etching times were required to dissolve most of the non-zircon components. Since HF attacks all heavy minerals except zircon (Sutton and Zimmerman, 1976), extended etching times should not negatively affect sample quality. Aliquots of the etched samples were examined under a binocular microscope to assess zircon purity. Neither this inspection nor cathodoluminescence imaging or LA-ICP-MS analyses showed any evidence of etching pits or changes in zircon morphology resulting from HF etching.
Furthermore, we did not add $H_2SO_4$, hence there is the possibility of accumulation of $SiF_6^{2-}$ anions in the solution, buffering the silicates and slowing their dissolution. Adding $H_2SO_4$ will break the hexafluoride anion and liberate $SiF_4$ as a gas that escapes from the reaction mixture. Finally, it is expected that the dissolution rate of volcanic glass (and, actually, any silicate glass), is faster compared to crystalline phases. Therefore, the required etching time also depends on the type of sample hosting the zircons.

- Comparison of zircon photoionisation cross-sections with quartz OSL components

Thank you for this critical comment. We recognize that the components fitted in zircon and quartz OSL decay curves may not be physically related. However, the aim of this paragraph was also to provide the reader with a benchmark regarding the optical resetting rates of zircon in relation to minerals more

commonly used in luminescence geochronology, such as quartz. Following the recommendation, we adapt the paragraph and focus on the comparison of photoionisation cross-sections, without relating to quartz OSL components.

- Shortening the discussion by removing the section on the dating range

As both reviewers ask for adapting the discussion to better link it to the experimental data presented in our study, we decide to remove section 4.2.4.

- Figure 7

Following the recommendation, we remove the lines indicating the quartz s1 and s3 components from Fig. 7 in the revised version of the manuscript. In addition, we create a plot of photoionisation cross-section versus component number for the two samples, as suggested, and show this plot in the supplement.

**References**

Amin, Y., Bull, R., Green, P., Durrani, S., 1983. Effect of radiation damage on the TL properties of zircon crystals. Third specialist seminar on TL and ESR dating.

Smith, B., 1988. Zircon from sediments: a combined OSL and TL auto-regenerative dating technique. Quaternary Science Reviews 7, 401-406.

Sutton, S., Zimmerman, D., 1976. Thermoluminescent dating using zircon grains from archaeological ceramics. Archaeometry 18, 125-134.

Templer, R., Smith, B., 1988. Auto-regenerative TL dating with zircon inclusions from fired materials. International Journal of Radiation Applications and Instrumentation. Part D. Nuclear Tracks and Radiation Measurements 14, 329-332.

Vaz, J.E., Senftle, F.E., 1971. Thermoluminescence study of the natural radiation damage in zircon. Journal of Geophysical Research 76, 2038-2050.